# Longitudinal trends in renal function among first time sugarcane harvesters in Guatemala

**Miranda Dally**[1,2]*, **Jaime Butler-Dawson**[1,2], **Alex Cruz**[3], **Lyndsay Krisher**[1], **Richard J. Johnson**[4], **Claudia Asensio**[3], **W. Daniel Pilloni**[3], **Edwin J. Asturias**[5,6,7], **Lee S. Newman**[1,2,7,8]

**1** Center for Health, Work, & Environment, Colorado School of Public Health, University of Colorado Anschutz Medical Campus, Aurora, Colorado, United States of America, **2** Department of Environmental and Occupational Health, Colorado School of Public Health, University of Colorado Anschutz Medical Campus, Aurora, Colorado, United States of America, **3** Pantaleon, Guatemala City, Guatemala, **4** Division of Renal Diseases and Hypertension, School of Medicine, University of Colorado Anschutz Medical Campus, Aurora, Colorado, United States of America, **5** Division of Pediatric Infectious Diseases, School of Medicine, University of Colorado, Anschutz Medical Campus, Aurora, Colorado, United States of America, **6** Center for Global Health, Colorado School of Public Health, University of Colorado Anschutz Medical Campus, Aurora, Colorado, United States of America, **7** Department of Epidemiology, Colorado School of Public Health, University of Colorado Anschutz Medical Campus, Aurora, Colorado, United States of America, **8** Division of Pulmonary Sciences and Critical Care Medicine, School of Medicine, University of Colorado Anschutz Medical Campus, Aurora, Colorado, United States of America

* miranda.dally@cuanschutz.edu

**Data Availability Statement:** All files to reproduce the findings in this manuscript are housed on Open Science Framework (https://osf.io) DOI 10.17605/OSF.IO/4YGSB.

## Abstract

### Introduction

Chronic kidney disease of unknown origin (CKDu) is an epidemic concentrated in agricultural communities in Central and South America, including young, male sugarcane harvesters. The purpose of this analysis is to understand early changes in kidney function among a cohort of first-time sugarcane harvesters and to determine risk factors for kidney function decline.

### Methods

Joint latent class mixed models were used to model sub-population kidney function trajectory over the course of 4 years (2012–2016). Probability weighted logistic regression was used to determine personal health, community, and individual behavior risk factors associated with sub-population assignment. Data analysis occurred in 2019.

### Results

Of 181 new workers median age 19 years old (IQR: 4), 39 (22%) were identified as having non-stable kidney function with an annual age-adjusted decline of estimated glomerular filtration rate (eGFR) of -1.0 ml/min per 1.73 m$^2$ (95% CI: -3.4, 1.3). Kidney function (OR: 0.96; 95% CI: 0.93, 0.98), mild hypertension (OR: 5.21; 95% CI: 2.14, 13.94), and having a local home of residence (OR: 7.12; 95% CI: 2.41, 26.02) prior to employment in sugarcane were associated with non-stable eGFR sub-population assignment.

**Funding:** This evaluation was supported in part by Pantaleon; the Chancellor, CU Anschutz; Centers for Disease Control and Prevention (CDC.gov) (U19 OH01127 to LSN) and National Institutes of Health (NIH.gov) (R21 ES028826 to LSN). Its contents are solely the responsibility of the authors and do not necessarily represent the official views of the Centers for Disease Control and Prevention, the National Institutes of Health, or the Department of Health and Human Services. The funders had no role in the evaluation design, data analysis, or data interpretation. Co-authors employed by the company (AC, CA, WDP) participated on the writing team. They provided details regarding work practices, but did not modify the results or conclusions of this report. The corresponding author had full access to all the raw data and had final responsibility for the decision to submit for publication. University of Colorado and Pantaleon are separate, independent organizations. University of Colorado employed appropriate research methods in keeping with academic freedom, based conclusions on critical analysis of the evidence and reported findings fully and objectively. The terms of this arrangement have been reviewed and approved by the University of Colorado in accordance with its conflict of interest policies.

**Competing interests:** This evaluation was partially supported by Pantaleon, a commercial founder. This does not alter our adherence to PLOS ONE policies on sharing data and material.

## Conclusions

Mild hypertension may be an early indicator of the development of CKDu. A better understanding of preexisting risk factors is needed to determine why individuals are entering the workforce with reduced kidney function and elevated blood pressure and increased risk of renal function decline.

## Introduction

Chronic kidney disease of unknown origin (CKDu) is an emerging epidemic concentrated in agricultural communities in Central and South America [1]. Unlike other forms of chronic kidney disease, CKDu is defined in the absence of traditional risk factors such as diabetes and hypertension [2, 3]. Characterized as a form of tubulointerstitial nephritis with varying degrees of fibrosis [4], clinical and epidemiologic studies have shown that young men are at the greatest risk for developing the disease with the age at diagnosis most often between 30 and 50 years [2].

Because the epidemic is concentrated in areas where a high percentage of the labor force conducts physically demanding manual labor [5] one of the leading hypotheses is that CKDu is caused, at least in part, by carrying out heavy work in hot climates while under a state of dehydration [1, 6–8]. Much of the research has been focused in either agricultural worker cohorts or community cohorts where a large percentage of the population is composed of current or former agricultural workers [2, 9–13]. These cross-sectional studies have implicated sugarcane harvesting and the number of years of work in agriculture as predictors of development of disease [10, 11]. However, some recent lines of evidence suggest that CKDu may also be occurring in communities among individuals who are not agricultural workers [14–17]. Recent studies suggest that there may be early indicators of renal damage among adolescents [18, 19], suggesting the possibility that there may be non-occupational contributors to the development of disease.

In order to better understand the possible combined contributions of individual health, community, and work-related risk factors to renal function decline, we conducted a four-year longitudinal analysis of apparently healthy, asymptomatic young men seeking their first job as sugarcane harvesters in southwest Guatemala. We hypothesized that there are multifactorial community, personal health, and lifestyle risk factors present prior to employment that contribute to observed changes in renal function over the course of their time employed in sugarcane.

## Methods

### Cohort

Workers from the region surrounding a sugarcane mill in southwest Guatemala (local workers), as well as migrant workers from other parts of Guatemala, are recruited annually by a large agribusiness. Each year, prior to the start of the 6-month sugarcane harvest, the agribusiness conducts a pre-employment screening which includes a medical exam (blood pressure, heart rate, height, and weight), questionnaire (demographic, lifestyle, medical, and occupational history questions), and venipuncture for measurement of serum creatinine of each individual seeking employment as a sugarcane harvester. Full details on recruitment and pre-employment screening are published elsewhere [20].

Subjects for our analysis were retrospectively selected from workers seeking employment as sugarcane harvesters between November 2012 and November 2015 at the agribusiness. To be included in this analysis, workers must have 1) reported never working a previous sugarcane harvest, whether at this company or another, 2) been hired and worked at least one week of the harvest after hiring date, and 3) have returned for pre-employment screening the subsequent year. Once this cohort was identified, the agribusiness used employment records to verify that these individuals had not previously worked for the company.

Data from the pre-employment screenings and information on the productivity of workers (to verify employment and attrition) were provided by the agribusiness. Data analysis occurred in 2019. As this was a secondary evaluation of de-identified data collected for business purposes, informed consent was not obtained. Ethics review and approval for our evaluation of these data was received from Colorado Multiple Institutional Review Board (COMIRB).

## Outcome

Creatinine measures were collected at the time of pre-employment screenings. Blood samples were sent to an independent, licensed clinical laboratory (Herrera Llerandi laboratory, Guatemala City, Guatemala). The Creatinine Jaffe Generation 2 method was used to determine serum creatinine. Further detail on the laboratory methods has been published elsewhere [21]. The primary outcome was the pattern of change in estimated glomerular filtration rate (eGFR) over a period of up to 4 years (2012–2016). Data on eGFR from 2016 were used for follow-up eGFR measurements only. To calculate the eGFR we used the Chronic Kidney Disease Epidemiology Collaboration (CKD-EPI) formula [22]. Race was considered "non-black" and all workers were male. From 2009 to 2015 it was the policy of the company to only hire workers with a serum creatinine of 1.45 mg/dL or less. The policy has since been updated (2017) to only hire cane cutters with an eGFR above 90 ml/min per 1.73 m$^2$.

## Covariates and potential risk factors

Baseline covariates were defined based on the first pre-employment screening of an individual. These included eGFR, age, body mass index (BMI), blood pressure, location of home residence, home drinking water source, alcohol consumption, and smoking status.

Pre-employment blood pressure measurements were taken by a licensed nurse following standard protocol. If there was doubt regarding the reading, a medical doctor in charge of pre-employment evaluation obtained a second measurement. The worker was notified of his result. We used the cut-off of systolic blood pressure ≥130 or diastolic blood pressure ≥ 80 to define mild hypertension in accordance with the 2017 American Academy of Cardiology guidelines [23].

## Statistical methods

Because kidney function at time of hire is related to attrition from the workforce [24] we used joint latent class mixed models to model the shape of longitudinal kidney function change while simultaneously modeling loss to follow-up [25]. In our analysis, time was treated as season (1–5 harvest seasons) in the linear mixed model. The longitudinal change in eGFR was modelled with a quadratic time trend with random terms for intercept and 2-degree polynomial time. Unconditional models were used to determine class-membership. Class-specific baseline risk functions were adjusted for continuous age. A full discussion of our modelling approach and final model specifications can be found in the Supporting Information. The R package "lcmm" was used [26].

**Sub-population assignment.**   Each individual was assigned a probability for sub-population assignment from the joint latent class mixed model. Annual change in eGFR for each sub-population was calculated using an age-adjusted Bayesian generalized linear multilevel model with random intercept for individual [27]. Age-adjusted associations between baseline covariates, as measured by their first pre-employment survey as described above, and probability of sub-population assignment were assessed using probability weighted logistic regression. Models were then further adjusted for baseline eGFR. For descriptive tables, individuals were assigned to the sub-population with which they had the highest probability. Comparisons between groups were done using Chi-squared, Fisher's Exact, or T-tests as appropriate. All analyses were conducted in R version 3.4.3 [28].

**Sensitivity analysis.**   To address the potential misclassification of individuals with mild hypertension we conducted a sensitivity analysis. We took random samples of 20% and 40% of the workers with mild hypertension and re-classified them as non-hypertensive. We re-ran the sub-population assignment analysis as described above with the updated datasets to assess the stability of the hypertension estimates.

## Results

### Cohort and follow-up

We identified 534 male individuals who were hired and worked at least one week as first time sugarcane harvester between 2012 and 2015. Of these, 181 (34%) returned the subsequent year for the next pre-employment screening, making up the analysis cohort (S1 Fig). Of the 181 workers with longitudinal measurements of eGFR, subsequent measures of eGFR were available for 100 workers for one year, 52 workers for two years, 26 workers for three years, and 3 workers for four years. A comparison between those selected and not selected for the cohort is provided in the Supporting Information (S1 Table).

### eGFR sub-populations

We identified two distinct sub-populations of first year sugarcane harvesters: those who slightly declined in kidney function over time (non-stable) and those who remained stable (Fig 1). Most workers fell into the stable category (78%; n = 142); however, almost a quarter (22%; n = 39) were identified in the non-stable category. The average annual change in eGFR for workers in the non-stable group was -1.0 ml/min per 1.73 $m^2$ (95% CI: -3.4, 1.3) compared to 0.3 ml/min per 1.73 $m^2$ (95% CI: -0.9, 1.5) for the stable group.

The two groups did significantly differ on baseline eGFR. While no workers in our analysis had baseline kidney dysfunction when hired (defined as eGFR < 60 ml/min per 1.73 $m^2$) [29], five workers assigned to the non-stable group had eGFR < 90 ml/min per 1.73 $m^2$ at the time of hire (Minimum: 84 ml/min per 1.73 $m^2$). Those in the non-stable group had on average a baseline eGFR 17 ml/min per 1.73 $m^2$ lower (95% CI: -21 ml/min per 1.73 $m^2$, -12 ml/min per 1.73 $m^2$) than those in the stable group (Table 1). They also tended to be older. The average baseline BMI for those in the non-stable group was higher than those in the stable group.

There were notable differences in the distribution of blood pressure between the non-stable and stable groups (Fig 2). A total of 85 workers (47%) in this analysis met the definition of mild hypertension. The proportion of individuals with mild hypertension differed significantly between groups, with 26 (67%) individuals in the non-stable eGFR group having mild hypertension compared to 59 (42%) in the stable eGFR group (p-value: 0.01).

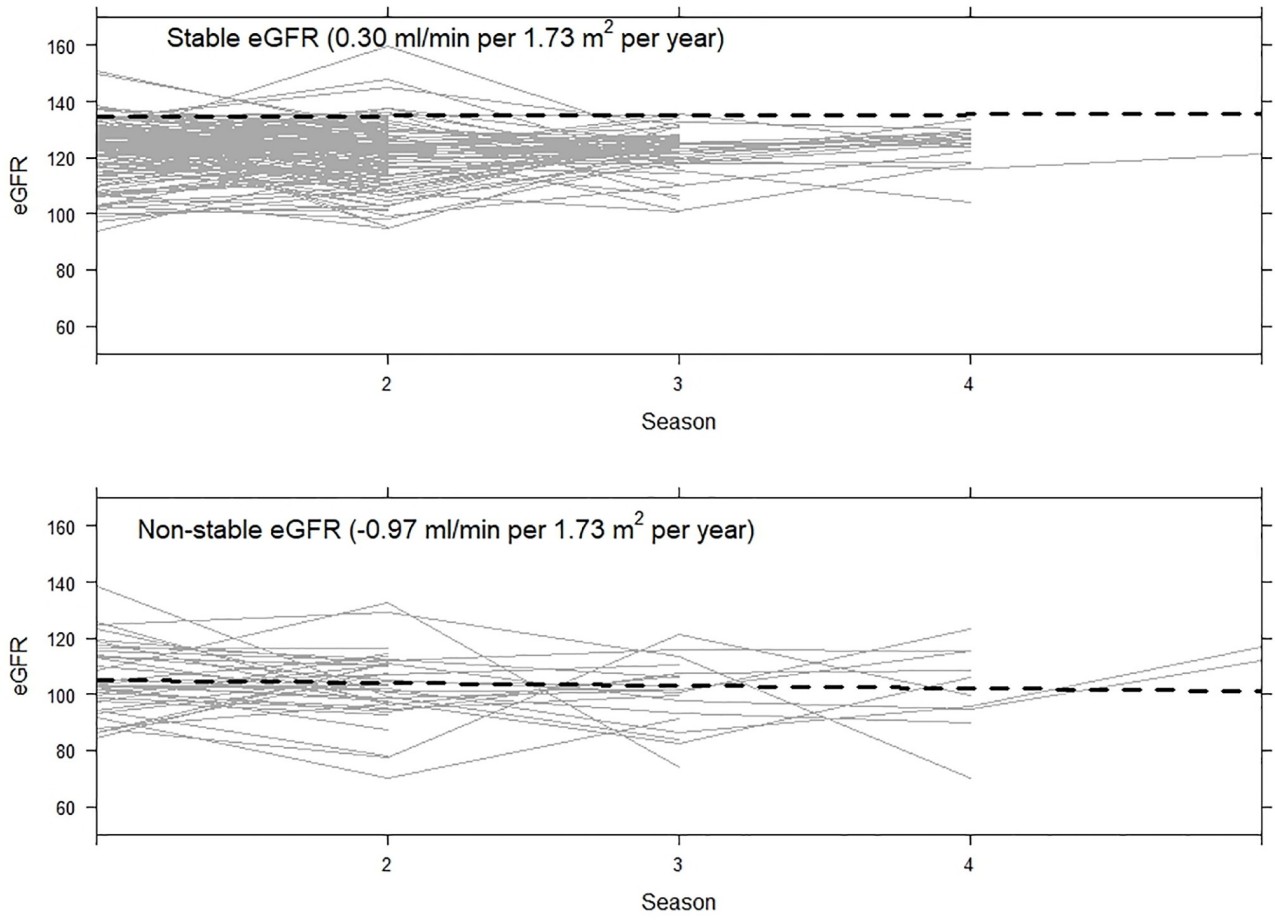

**Fig 1. Individual longitudinal eGFR patterns stratified by assigned sub-population.**

### Adjusted associations with non-stable kidney function

In age-adjusted probability weighted models, baseline eGFR and baseline creatinine were associated with non-stable eGFR sub-population assignment (Table 2). Odds of assignment into the non-stable group decreased as baseline eGFR increased (OR: 0.96; 95% CI: 0.93, 0.98). In models that adjusted both for age and baseline eGFR, new workers with a local home of residence had 7.12 times the odds of being in the non-stable subgroup (95% CI: 2.41, 26.02) compared to those from other regions. Those with a well water source at home had 6.1 times the odds of being in the non-stable subgroup (95% CI: 1.80, 26.82) compared to all other sources of home water. Water source was highly correlated with region, with 60% of those with a local home of residence using well water compared to 13% from other regions (p-value: <0.001). Notably, the individuals with mild hypertension at the time of hire had 5.2 times the odds of assignment to the non-stable group (95% CI: 2.14, 13.94) compared to those with normal blood pressure. In a sensitivity analysis with 20% of the mild hypertensive workers randomly re-assigned to non-hypertensive, the results held (OR: 6.00; 95% CI: 2.24, 18.98; p-value: 0.002). Estimates were unstable in the extreme case of 40% of mild hypertensive workers randomly re-classified (OR: 0.54; 95%CI: 0.22, 1.33; p-value 0.17). A summary of all sensitivity analysis results is presented in S2 Table.

**Table 1. Baseline covariates of new sugarcane workers by assigned sub-population in mean (SD) or n (%).** Data collected from workers seeking first time work as sugarcane harvesters in Guatemala from 2012–2015.

| Baseline Characteristic | Non-stable Kidney Function (n = 39) | Stable Kidney Function (n = 142) | p-value |
|---|---|---|---|
| Creatinine, mg/dL | 0.97 (0.12) | 0.86 (0.10) | <0.001 |
| eGFR, ml/min per 1.73 m$^2$ | 105.24 (12.43) | 123.33 (10.11) | <0.001 |
| Age, years | 28 (7) | 20 (2) | <0.001 |
| BMI, kg/m$^2$ | 23.75 (2.70) | 22.60 (2.15) | 0.005 |
| Systolic, mmHg | 111 (10) | 110 (12) | 0.686 |
| Diastolic, mmHg | 76 (7) | 73 (8) | 0.050 |
| Hypertension[a] | 26 (67%) | 59 (42%) | 0.009 |
| Stage 1 | 25 (64%) | 49 (35%) | 0.002 |
| Stage 2 | 1 (3%) | 10 (7%) | 0.461 |
| Local home of residence (vs. migrant) | 16 (41%) | 46 (32%) | 0.314 |
| Well water source | 11 (28%) | 42 (30%) | 0.868 |
| Consumes alcohol | 2 (5%) | 7 (5%) | 0.868 |
| Current or former smoker | 0 (0%) | 12 (9%) | 0.072 |
| Weeks worked first year | 26 (1) | 26 (2) | 0.500 |

[a]Stage 1 hypertension defined as systolic blood pressure ≥130 or diastolic blood pressure ≥ 80; Stage 2 hypertension defined as systolic blood pressure ≥140 or diastolic blood pressure ≥ 90.

## Discussion

To our knowledge this is the first longitudinal analysis examining kidney function decline in a young, new worker population at a risk of developing CKDu. This analysis established that among newly hired sugarcane cutters, between 20% and 25% are expected to experience non-stable eGFR, with an average decline in eGFR at a rate of 1 ml/min per 1.73 m$^2$ per year after adjusting for age. Workers who entered the workforce with lower levels of kidney function and mild hypertension were at a greater risk for experiencing declines in kidney function. Coupled with home of residence being an additional predictive risk factor for decline in kidney function suggests early exposures and community level factors play a role in the development of CKDu.

Mild hypertension at time of hire increased the odds of non-stable eGFR group assignment 5-fold. In addition, elevations of continuous diastolic blood pressure were also suggestive of non-stable eGFR in our age- and baseline eGFR-adjusted models. This is a notable finding because CKDu has been defined in absence of hypertension (systolic blood pressure ≥140 or diastolic blood pressure ≥ 90) but has not been examined in relationship to mild elevations of blood pressure. Our findings show that the non-stable group had both a higher prevalence of mild hypertension as well as lower baseline eGFR. Systemic arterial hypertension is almost

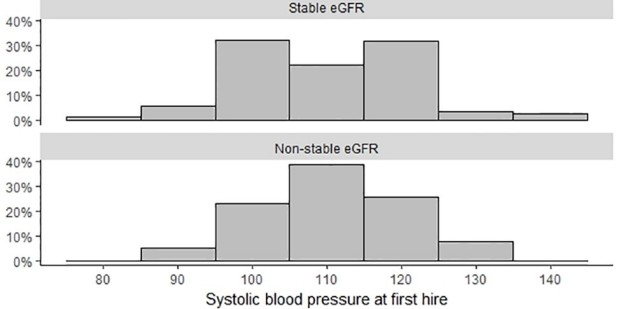 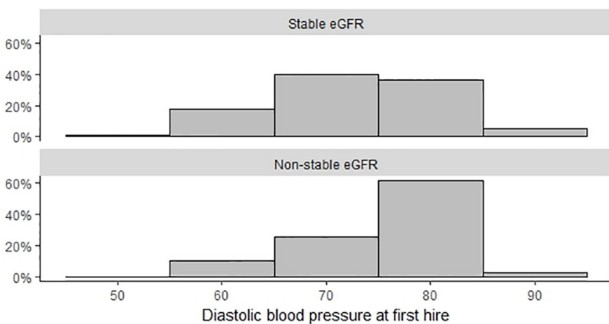

**Fig 2. Blood pressure measurements at the time of first hire for first year sugarcane harvesters.**

**Table 2. Age-adjusted and age- and baseline eGFR-adjusted Odds Ratio (95% CI) of non-stable eGFR sub-population assignment.** Data collected from workers seeking first time work as sugarcane harvesters in Guatemala from 2012–2015.

| | Age-adjusted OR for Non-stable | p-value | Age- and baseline eGFR adjusted OR for Non-stable | p-value |
|---|---|---|---|---|
| Baseline Creatinine (per 0.10 mg/dL) | 1.56 (1.15, 2.15) | 0.005 | | |
| Baseline eGFR, ml/min per 1.73 m2 | 0.96 (0.93, 0.98) | 0.004 | | |
| Baseline BMI, kg/m$^2$ | 0.98 (0.81, 1.22) | 0.864 | 0.84 (0.67, 1.05) | 0.122 |
| Baseline Systolic (per 10 mmHg) | 0.96 (0.64, 1.44) | 0.850 | 0.95 (0.66, 1.36) | 0.769 |
| Baseline Diastolic (per 10 mmHg) | 1.45 (0.80, 2.64) | 0.225 | 1.72 (0.97, 3.10) | 0.066 |
| Baseline Hypertension[a] | 2.80 (1.20, 6.74) | 0.020 | 5.21 (2.14, 13.94) | 0.001 |
| Local home of residence vs. migrant | 3.95 (1.40, 13.42) | 0.016 | 7.12 (2.41, 26.02) | 0.001 |
| Well water source | 3.48 (1.19, 12.94) | 0.038 | 6.07 (1.80, 26.82) | 0.009 |
| Weeks worked | 1.19 (0.70, 1.91) | 0.488 | 1.01 (0.57, 1.62) | 0.964 |

[a]Hypertension defined as systolic blood pressure ≥130 or diastolic blood pressure ≥ 80.

always associated with, and likely driven by, subtle renal disease [30]. This, coupled with our findings presented here, suggest mild hypertension as an early indicator of the development of CKDu. Blood pressure screening in conjunction with renal function screenings should become a routine part of health monitoring for workers and community members at risk of CKDu, specifically those performing intense labor in hot environments. In addition, more research is warranted on the role of elevations of blood pressure, longitudinal changes in blood pressure, and the development of CKDu.

Our analysis showed that location of home of residence was associated with non-stable eGFR group assignment after adjusting for age and baseline eGFR. Workers living locally come from the communities surrounding the mill, which are at lower altitudes and closer to the coast than the seasonal migrant workers who generally are from the highlands, which are more mountainous, and at a higher altitude. Previous studies have shown that lower altitude and coastal home of residence are risk factors for CKDu [16, 31, 32] potentially due to differences in climatic factors and job history differences between these regions. While we are confident that the workers included in this analysis had never worked in sugarcane, we did not account for their job histories. There is the potential that the individuals evaluated here have had previous employment in other agricultural settings or participated in subsistence agriculture, which can differ by region, potentially explaining some of the differences we found between local and migrant workers. Limited research has been conducted to identify other differences in CKDu risk factors between local and migrant communities, but possible explanations could include differences in diets [33], airborne exposures [34], or water sources [35] which we showed to increase non-stable group assignment 6-fold and was highly associated with home of residence. Understanding differences in childhood and adolescent exposures may help determine why some young men enter the workforce with lower than expected eGFR and with elevated blood pressure.

This analysis carries several strengths. The models we employed help overcome common obstacles in most studies of CKDu, mainly the need for repeated measurements of eGFR, assumptions of linear trends in eGFR decline, and loss to follow-up. First, we had measurements on 181 first time sugarcane harvesters for up to four years, longer than any study we are aware of in the CKDu literature. Second, the joint linear mixed model allowed us to be flexible with the functional form of time when identifying sub-populations of renal function change. Third, informative loss to follow up in longitudinal worker cohorts is an issue, because workers with lower kidney function at the start of the harvest season are less likely to return to work [24]. Using joint linear mixed models allowed us to leverage this information.

## Limitations

Despite these strengths, this analysis does face some limitations. We were limited in our ability to determine factors associated with a lower baseline eGFR, higher baseline blood pressure, and factors that differentiate local and migrant workers. Because these data were collected for hiring purposes, we were limited in our ability to examine additional hypothesized risk factors such as heat exposure [6, 11], use of pesticides [36, 37], and exposure to heavy metals [34, 38]. Due to the structure of our data, we were unable to assess the differences in recurrent dehydration [11], NSAID use [20], and other factors previously implicated in the development of the disease between the two identified groups. There is a possibility of nondifferential misclassification of survey variables including smoking status and alcohol intake. Due to the low number of respondents to questions regarding smoking and alcohol consumption we were unable to test the relationship with group assignment. We would be remiss not to mention that hypertension is clinically defined by elevated measurements at two time points, however we were limited in our classification to the single measurement taken at the time of hire. Finally, we lost many individuals to follow-up, limiting how many individuals for whom we had more than two time points. While we used a modeling approach that leveraged this information, it cannot be ignored that there are both observed and unobservable differences between those who choose to seek employment in subsequent years. Additionally, as loss to follow-up is associated with reduced kidney function [24], our analysis may underestimate the true rate of renal function decline as well as the association between renal function decline and mild-elevations in blood pressure.

## Conclusions

To our knowledge this is the first cohort that examines the course of renal function change in newly hired, first time sugarcane harvesters who are at risk of CKDu. We identified nearly a quarter of the apparently healthy young men who experience on average a decline in eGFR of 1 ml/min per 1.73 $m^2$ per year. Risk factors for this decline were decreased baseline eGFR, elevated baseline blood pressure, and residing locally in communities near the mill. A better understanding of non-occupational exposures and individual risk factors are needed to determine why these individuals are entering the workforce at higher risk. Improved surveillance practices of blood pressure and kidney function measurement are needed in the workplace in order to identify individuals at risk of kidney function decline. Workplaces should implement eGFR hiring cutoffs at 90 ml/min per 1.73 $m^2$ as our study shows that all workers who had an eGFR < 90 ml/min per 1.73 $m^2$ at the time of hire were assigned to the non-stable group. Studies of the possible mechanisms by which the combination of community, individual, and workplace risk factors contribute to renal function decline are needed.

## Supporting information

**S1 Fig. The analysis cohort was drawn from workers seeking employment as first time sugarcane harvesters between November 2012 and November 2015.** Study flow showing timeline along with the number of new workers screened and the number returning each subsequent year. (TIF)

**S1 File. Modelling approach and model selection.** (DOCX)

**S2 File. Spanish versions of the pre-employment data collection form created and used by Pantaleon.** (DOCX)

**S3 File. English versions of the pre-employment data collection form created and used by Pantaleon.**
(DOCX)

**S1 Table. Baseline characteristics between those identified as first year sugarcane harvesters who sought subsequent employment (included in the analysis) and those who only worked a single harvest (excluded from the analysis).**
(DOCX)

**S2 Table. Age-adjusted and age- and baseline eGFR-adjusted associations of non-stable eGFR subgroup assignment (referent: Stable) by hypertension with a random sample of 20% and 40% of hypertensive workers reclassified as non-hypertensive.** Presented as Odds Ratio (95% Confidence Interval).
(DOCX)

## Acknowledgments

We would like to thank the men and women employed by Pantaleon who worked tirelessly to ensure the accuracy of the data used in this evaluation. We would also like to thank Alicen Nelson, MD, MPH for her pilot work in assessing the pre-employment variables included in this analysis.

## Author Contributions

**Conceptualization:** Miranda Dally, Jaime Butler-Dawson, Lyndsay Krisher, Lee S. Newman.

**Data curation:** Miranda Dally, Jaime Butler-Dawson.

**Formal analysis:** Miranda Dally.

**Funding acquisition:** Lee S. Newman.

**Project administration:** Lyndsay Krisher.

**Resources:** Alex Cruz, Claudia Asensio, W. Daniel Pilloni.

**Supervision:** Lee S. Newman.

**Visualization:** Miranda Dally.

**Writing – original draft:** Miranda Dally, Jaime Butler-Dawson, Lyndsay Krisher, Lee S. Newman.

**Writing – review & editing:** Alex Cruz, Richard J. Johnson, Claudia Asensio, W. Daniel Pilloni, Edwin J. Asturias.

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
