## [Decision Letter · Decision Letter 0]

21 Jan 2020

PONE-D-19-32490

Longitudinal trends in renal function among first time sugarcane harvesters in Guatemala

PLOS ONE

Dear Ms. Dally,

Thank you for submitting your manuscript to PLOS ONE. After careful consideration, we feel that it has merit but does not fully meet PLOS ONE’s publication criteria as it currently stands. Therefore, we invite you to submit a revised version of the manuscript that addresses the points raised during the review process.

We would appreciate receiving your revised manuscript by Mar 06 2020 11:59PM. To enhance the reproducibility of your results, we recommend that if applicable you deposit your laboratory protocols in protocols.io, where a protocol can be assigned its own identifier (DOI) such that it can be cited independently in the future. For instructions see: http://journals.plos.org/plosone/s/submission-guidelines#loc-laboratory-protocols

We look forward to receiving your revised manuscript.

Kind regards,

Davide Bolignano, MD

Academic Editor

PLOS ONE

Journal Requirements:

This evaluation was supported in part by Pantaleon; the Chancellor, CU Anschutz; Centers for Disease Control and Prevention (CDC.gov) (U19 OH01127 to LSN) and National Institutes of Health (NIH.gov) (R21 ES028826 to LSN). Its contents are solely the responsibility of the authors and do not necessarily represent the official views of the Centers for Disease Control and Prevention, the National Institutes of Health, or the Department of Health and Human Services.

The funders had no role in the evaluation design, data analysis, or data interpretation. Co-authors employed by the company (AC, CA, WDP) participated on the writing team. They provided details regarding work practices, but did not modify the results or conclusions of this report. The corresponding author had full access to all the raw data and had final responsibility for the decision to submit for publication. University of Colorado and Pantaleon are separate, independent organizations. University of Colorado employed appropriate research methods in keeping with academic freedom, based conclusions on critical analysis of the evidence and reported findings fully and objectively. The terms of this arrangement have been reviewed and approved by the University of Colorado in accordance with its conflict of interest policies.

We note that you received funding from a commercial source: Pantaleon

Reviewers' comments:

Reviewer's Responses to Questions

**Comments to the Author**

1. Is the manuscript technically sound, and do the data support the conclusions?

Reviewer #1: Yes

2. Has the statistical analysis been performed appropriately and rigorously? 

Reviewer #1: Yes

3. Have the authors made all data underlying the findings in their manuscript fully available?

Reviewer #1: Yes

4. Is the manuscript presented in an intelligible fashion and written in standard English?

Reviewer #1: Yes

5. Review Comments to the Author

Reviewer #1: In Mesoamerica, chronic kidney disease of un-known origin (CKDu) is an important public health problem and this is because of several reasons: 1) This pidemic entity affects young and middle-aged men, resulting in high social-economic costs for health system. 2) Many causal risk factors have been hypothesized, but pathogenesis is still unexplained, therefore further studies are necessary. 3) Patients are often not aware of the disease, until CKDu reach final stages. Moreover, most patients have not the chance for renal replacement therapy and CKDu has become a leading cause of mortality. In this article Authors discuss on kidney disease development, trying to find new causes. One of the underlying factors is the presence of mild hypertension among this cohort, the unexpected prevalence of this latter, even among young men, raises new questions.

Comments:

- According to the esponential relationship between eGFR and creatinine, normal values of this latter can be related to an eGFR range between 50 and 180 ml/min. In other words, initial changes in eGFR determine little variations of creatinine, while pathological filtrates are associated to greater alterations of creatinine. Thus, resulting in a low sensitivity for this relationship in the earlier states of renal failure. The high current cut-off adopted by Company (serum creatinine level of 1.45 mg/dL or less) determines enrollment of individuals with impaired renal function. The Authors may concern about the need to decrease this value.

- The baseline covariates, defined at the screening visit, included: eGFR, age, body mass index (BMI), blood pressure, location of home residence, home drinking water source, alcohol consumption, and smoking status. However, potentially implicated factors have not been included, such as: daily water intake amount (dehydration could be contributing to heat stress), use of NSAIDs (very common among this populations). These latter should be mentioned in Limitations.

- Regarding paragraph on pre-employment blood pressure measurements, it seems to be much descriptive. They should be summarized in some extents.

- The Authors found that mild hypertension (≥130/80 mmHg) could be one of the underlying causes of CKDu. This evidence is remarkable because other Authors had already depicted the linear relationship between blood pressure and kidney damage, even starting from normal values. However, the authors should mention that in non-adjusted and in age- and baseline eGFR- adjusted OR for non-stable eGFR, the study does not reach statistical significance on highly suspect causes implicated in CKDu such as: well water source, current or past smoking, alcohol consumption.

- Patients loss at the follow up can cause bias, workers may have been absent due to the kidney disease. It should be mentioned that loss can lead to underestimation of renal function decline, influencing relationship between eGFR and risk factor.

6. PLOS authors have the option to publish the peer review history of their article (what does this mean?). If published, this will include your full peer review and any attached files.

Reviewer #1: Yes: Michele Provenzano

---

## [Author Response · Author response to Decision Letter 0]

3 Feb 2020

Dear Dr. Bolignano, 

We would like to thank Dr. Provenzano for taking the time to provide feedback on our manuscript, Longitudinal trends in renal function among first time sugarcane harvesters in Guatemala. Below, in bold, we have addressed each point brought up by Dr. Provenzano as well as editorial concerns.

Reviewer #1: In Mesoamerica, chronic kidney disease of un-known origin (CKDu) is an important public health problem and this is because of several reasons: 1) This pidemic entity affects young and middle-aged men, resulting in high social-economic costs for health system. 2) Many causal risk factors have been hypothesized, but pathogenesis is still unexplained, therefore further studies are necessary. 3) Patients are often not aware of the disease, until CKDu reach final stages. Moreover, most patients have not the chance for renal replacement therapy and CKDu has become a leading cause of mortality. In this article Authors discuss on kidney disease development, trying to find new causes. One of the underlying factors is the presence of mild hypertension among this cohort, the unexpected prevalence of this latter, even among young men, raises new questions.

Comments:

- According to the esponential relationship between eGFR and creatinine, normal values of this latter can be related to an eGFR range between 50 and 180 ml/min. In other words, initial changes in eGFR determine little variations of creatinine, while pathological filtrates are associated to greater alterations of creatinine. Thus, resulting in a low sensitivity for this relationship in the earlier states of renal failure. The high current cut-off adopted by Company (serum creatinine level of 1.45 mg/dL or less) determines enrollment of individuals with impaired renal function. The Authors may concern about the need to decrease this value.

Response: We agree that use of a creatinine of 1.45 mg/dL as a cutoff for hiring is too lax. In 2017, in response to our recommendations, the company changed the hiring cutoff to an eGFR above 90 ml/min per 1.73 m2. Despite the lenient hiring value during the time of our cohort construction, no worker in our analysis had a serum creatinine level above 1.20 mg/dL. The lowest observed baseline eGFR in our study was 84 ml/min per 1.73 m2. There were 5 individuals with a baseline eGFR below 90 ml/min per 1.73 m2 included in our analysis. We have updated the results section to mention this. We have also updated the conclusions to emphasize the need for a strict hiring cut-off. 

- The baseline covariates, defined at the screening visit, included: eGFR, age, body mass index (BMI), blood pressure, location of home residence, home drinking water source, alcohol consumption, and smoking status. However, potentially implicated factors have not been included, such as: daily water intake amount (dehydration could be contributing to heat stress), use of NSAIDs (very common among this populations). These latter should be mentioned in Limitations.

Response: We agree that a limitation of our study was our inability to assess factors previously implicated in the development of CKDu. We have updated our limitations section to more clearly reflect this. 

- Regarding paragraph on pre-employment blood pressure measurements, it seems to be much descriptive. They should be summarized in some extents.

Response: Thank you, we agree that we erred on the side of providing too much information and have summarized the blood pressure collection methodology. 

- The Authors found that mild hypertension (≥130/80 mmHg) could be one of the underlying causes of CKDu. This evidence is remarkable because other Authors had already depicted the linear relationship between blood pressure and kidney damage, even starting from normal values. However, the authors should mention that in non-adjusted and in age- and baseline eGFR- adjusted OR for non-stable eGFR, the study does not reach statistical significance on highly suspect causes implicated in CKDu such as: well water source, current or past smoking, alcohol consumption.

Response: Thank you for pointing out this concern to us. We have updated the limitation section to mention our inability to test smoking and alcohol consumption. We have updated the results and discussion to mention an association with well water source. 

- Patients loss at the follow up can cause bias, workers may have been absent due to the kidney disease. It should be mentioned that loss can lead to underestimation of renal function decline, influencing relationship between eGFR and risk factor.

Response: We agree that loss to follow-up and the “healthy worker effect” is a concern for research examining kidney function decline in worker cohorts. We have updated our limitations section to emphasize this. 

 Journal Requirements:

Response: We have updated the manuscript and supporting documents to meet the style requirements and file naming. 

Response: We have included both a Spanish and English copy of the pre-employment survey that was developed and administered by the business in the supporting information. Full specifications on how we implemented the joint latent class models is provided in the supporting information. 

This evaluation was supported in part by Pantaleon; the Chancellor, CU Anschutz; Centers for Disease Control and Prevention (CDC.gov) (U19 OH01127 to LSN) and National Institutes of Health (NIH.gov) (R21 ES028826 to LSN). Its contents are solely the responsibility of the authors and do not necessarily represent the official views of the Centers for Disease Control and Prevention, the National Institutes of Health, or the Department of Health and Human Services.

The funders had no role in the evaluation design, data analysis, or data interpretation. Co-authors employed by the company (AC, CA, WDP) participated on the writing team. They provided details regarding work practices, but did not modify the results or conclusions of this report. The corresponding author had full access to all the raw data and had final responsibility for the decision to submit for publication. University of Colorado and Pantaleon are separate, independent organizations. University of Colorado employed appropriate research methods in keeping with academic freedom, based conclusions on critical analysis of the evidence and reported findings fully and objectively. The terms of this arrangement have been reviewed and approved by the University of Colorado in accordance with its conflict of interest policies.

We note that you received funding from a commercial source: Pantaleon

Response: We have updated the Competing Interests Statement to reflect that our affiliation with Pantaleon does not alter our adherence to PLOS ONE policies on sharing data and materials and have included this in our cover letter.

Response: We have no changes to address currently.

---

## [Decision Letter · Decision Letter 1]

6 Feb 2020

Longitudinal trends in renal function among first time sugarcane harvesters in Guatemala

PONE-D-19-32490R1

Dear Dr. Dally,

We are pleased to inform you that your manuscript has been judged scientifically suitable for publication and will be formally accepted for publication once it complies with all outstanding technical requirements.

With kind regards,

Davide Bolignano, MD

Academic Editor

PLOS ONE

Additional Editor Comments (optional):

Reviewers' comments:

Reviewer's Responses to Questions

**Comments to the Author**

1. If the authors have adequately addressed your comments raised in a previous round of review and you feel that this manuscript is now acceptable for publication, you may indicate that here to bypass the “Comments to the Author” section, enter your conflict of interest statement in the “Confidential to Editor” section, and submit your "Accept" recommendation.

Reviewer #1: All comments have been addressed

2. Is the manuscript technically sound, and do the data support the conclusions?

Reviewer #1: Yes

3. Has the statistical analysis been performed appropriately and rigorously? 

Reviewer #1: Yes

4. Have the authors made all data underlying the findings in their manuscript fully available?

Reviewer #1: Yes

5. Is the manuscript presented in an intelligible fashion and written in standard English?

Reviewer #1: Yes

6. Review Comments to the Author

Reviewer #1: (No Response)

7. PLOS authors have the option to publish the peer review history of their article (what does this mean?). If published, this will include your full peer review and any attached files.

Reviewer #1: Yes: Michele Provenzano

---

## [Editor Report · Acceptance letter]

27 Feb 2020

PONE-D-19-32490R1 

Longitudinal trends in renal function among first time sugarcane harvesters in Guatemala 

Dear Dr. Dally:

I am pleased to inform you that your manuscript has been deemed suitable for publication in PLOS ONE. Congratulations! Your manuscript is now with our production department. 

With kind regards,

on behalf of

Dr. Davide Bolignano 

Academic Editor

PLOS ONE